# Morphology of the Female Reproductive System of the Soybean Thrips, *Neohydatothrips variabilis* (Beach, 1896) (Thysanoptera: Thripidae)

**DOI:** 10.3390/insects13070566

**Published:** 2022-06-23

**Authors:** Asifa Hameed, Jonah M. Ulmer, Istvan Miko, Cristina Rosa, Edwin G. Rajotte

**Affiliations:** 1Entomology, The Pennsylvania State University, 501 ASI, University Park, State College, PA 16802, USA; uvu@psu.edu; 2Entomology, State Museum of Natural History Stuttgart, Rosenstein 1, 70191 Stuttgart, Germany; jonah.ulmer@gmail.com; 3Biological Sciences, University of New Hampshire Durham, Spaulding Hall Rm 266, Durham, NH 03824, USA; Istvan.miko@gmail.com; 4Plant Pathology and Environmental Microbiology, Pennsylvania State University, University Park, State College, PA 16802, USA; czr2@gmail.com

**Keywords:** pest management, orthotospovirus, histology, vector thrips, agriculture

## Abstract

**Simple Summary:**

Soybean thrips are an important vector of Soybean vein necrosis virus disease, found in all soybean-growing regions of the United States and Canada. The disease reduces the seed oil content and fatty acid profiles in infected plants. It is important to know the morphology of the female reproductive system of soybean thrips to understand the egg-laying mechanism and develop strategies for virus vector management. For this purpose, we used fuchsin staining, paraffin-based histology, dissections, serial block-face scanning electron microscopy and confocal laser scanning microscopy techniques to understand the structure and motorization of the internal and external genitalia of soybean thrips. We also investigated the egg-laying behavior of soybean thrips. The female reproductive system is composed of two ovaries, an oviduct, an accessory gland, an appendage gland, a common oviduct and a vagina. Female soybean thrips lay eggs in the parenchymatous tissues near the veins of the leaves. The appendage gland likely secretes lubrication to facilitate the movement of eggs through the external genitalia. Seven muscles support the movement of eggs from the ovaries to the leaf surface. The anatomy and probable role of each muscle is also described.

**Abstract:**

Soybean thrips (*Neohydatothrips variabilis*) are an important phytophagous vector of the widely recognized Soybean vein necrosis orthotospovirus (SVNV). Understanding the egg-laying behavior of these thrips could aid in developing strategies for the management of the vector and virus. In this study, we described the egg-laying behavior of *N. variabilis* and reconstructed the three-dimensional morphology of the female terminalia by using serial block-face scanning electron microscopy (SBFSEM) and confocal laser scanning microscopy (CLSM). The female reproductive system consists of two panoistic ovaries consisting of eight ovarioles. The appendage gland is connected to the ovaries by two muscles, and to the body wall by a single muscle. The spermatheca is connected to the eighth tergum through four branched muscles, to the basivalvulae of the ovipositor by one muscle and to the vagina by a single muscle. The external genitalia are operated by seven muscles. The movement of the eggs inside the ovipositor is achieved by the back and forth “rocking” movement of the first valvulae and valvifer. Eggs are deposited into the parenchymatous tissue alongside leaf veins. To the best of our knowledge, this is the first study describing the internal and external genitalia of *N. variabilis*.

## 1. Introduction

Thrips belonging to the family Thripidae, order Thysanoptera, are economically important phytophagous and phytosanitary pests [1,2] because of the damage caused by oviposition, feeding and virus transmission [3]. Among 290 genera and around 2400 species of Thripidae [4], a few species, viz., *Frankliniella occidentalis* (Pergande), *Thrips palmi* (Karny), *Thrips tabaci* (Lindeman), *Scirtothrips dorsalis* (Hood) and *Neohydatothrips variabilis* (Beach) are important pests. Soybean thrips (*Neohydatothrips variabilis* (Beach, 1896)) have gained importance as vectors in recent years since the emergence of Soybean vein necrosis virus (SVNV) in the United States [5,6,7,8] and Canada [9]. Although SVNV’s presence has been reported in Egypt [10,11] as well, the presence of *N. variabilis* is still not confirmed. The primary vector of this virus is *N. variabilis* (Beach) (Thysanoptera: Thripidae), which is prevalent in every soybean-growing state in the United States [12] where SVNV’s presence has been reported [6,7,8]. Soybean vein necrosis virus is an important vector and seed-transmitted virus which decreases the seeds qualitative parameters including oil content and the fatty acid profile [7,13]. With SVNV continuing to affect soybean plants [14], there is a need to find ways to combat the spread of this virus and its vector. Understanding the morphology of the female terminalia of *N. variabilis*, as well as the egg -laying mechanism, could help to develop new strategies for managing both the vector and the disease.

Phytophagous thrips species are capable of reproducing parthenogenetically, allowing them to rapidly expand their population density [15]. Thysanoptera have haplodiploid sex determination, where haploid males develop from unfertilized eggs (arrhenotoky) and diploid females develop from fertilized eggs [16]. Due to their parthenogenetic reproduction, only female thrips are needed to establish populations in new areas, making thrips that act as phytophagous vectors a major threat to agriculture [17,18].

Understanding the structures of the female reproductive system, including ovipositor morphology, is also important for understanding the evolution of thrips and the interactions between thrips and their host plants [19]. Heming [20] documented that ovarioles in Thysanoptera are neopanoistic, modified from polytrophic ovarioles in Psocopteroid ancestors, which might be a modification for tiny size. Haplodiploidy is made possible because male thrips have haploid germline cells [20].

Female ovipositor morphology differs between the two suborders of thrips: whereas the members of Terebrantia have a well-developed ovipositor, members of the other order, Tubulifera, have an eversible chute like ovipositor that lacks teeth [21,22,23,24]. The specialized ovipositor structure in Terebrantia likely evolved during the Permian period (250–300 million years ago), when Terebrantia and Tubulifera split [25]. The terebrantian ovipositor is strong enough to pierce the plant leaf cuticle and place eggs inside the leaves where they are more protected from desiccation, predation and parasitism [26].

Despite their importance in sex determination and vector transmission, the morphology of the female genitalia has been only seldom described in Thysanoptera, and most of these descriptions are restricted to skeletal elements [27,28]. The only publication with a thorough description of the skeletomuscular and glandular system of female thrips’ terminalia was published in German almost 50 years ago [29]. Phylogenetic studies of Thysanoptera based upon 18S ribosomal DNA and 28S ribosomal DNA through maximum parsimony, maximum likelihood and Bayesian analysis revealed a the monophyletic Tubulifera and Terebrantia [30]. *Thrips physapus* L., the subject of Bode’s paper, and *N. variabilis*, our study organism, represent a single clade.

The objective of this study was to provide a detailed description of the oviposition behavior and the anatomy of the female reproductive system of *N. variabilis* by using fuchsin staining, paraffin-based histology, dissections, confocal laser scanning microscopy (CLSM) and serial block-face scanning electron microscopy (SBFSEM) to compare its features with that of *Thrips physapus*. Besides its putative evolutionary significance, understanding the female reproductive structures and the mechanism of oviposition may be useful in pest management programs to help develop new ways of managing SVNV.

## 2. Materials and Methods

### 2.1. Specimens

*N. variabilis* specimens (SVNV-infected) were collected from the soybean fields at the Russell E. Larson Agricultural Research Farm in State College, PA (40.71540N, 77.94290W and 200 m above sea level) during June–August 2016. Specimens were kept in the laboratory on soybean plants enclosed in insect-rearing cages (L24.5 × W24.5 × H24.5 cm) and placed in an environmental growth chamber (temperature 25 + 2 °C, humidity 80%, L:D 14:10) (Conviron, Winnipeg Manitoba, Canada). The specimens were identified to the species level [31]. Species identifications were confirmed by sending representative specimens to the national thrips specialist, Cheryle O’ Donnell, at the USDA APHIS PPQ NIS Systematic Entomology Laboratory in Baltimore, MA.

### 2.2. Egg-Laying Behavior

#### 2.2.1. Fuchsin Staining of Plant Tissues

To study the position structure of the eggs inside the leaf, soybean leaves were stained using the protocol established by Backus et al. [32]. Briefly, the eggs were stained in a McBryde [33] solution consisting of 0.2% acid fuchsin in 95% ethanol and glacial acetic acid (1:1 vol/vol), and incubated at room temperature overnight. A second solution was prepared as a clearing agent, which consisted of distilled water, 99% glycerin and 85% lactic acid (1:1:1 vol/vol/vol). After 20–24 h, the leaves were dipped in the clearing agent, removed and again washed in the clearing agent. 

#### 2.2.2. Histology of Egg-Containing Leaves

To visualize the eggs, the leaves were placed in an autoclave for 15 to 20 min (93–120 °C at 5.0 to 5.4 mm Hg). The tissue was then cooled and placed under the dissecting microscope for observation. For permanent mounting of the eggs inside the leaves, the leaves were paraffin-waxed, sectioned with the microtome at 4 µm and then stained. Slides were observed under the dissecting microscope. 

### 2.3. Morphology of Female Terminalia

#### 2.3.1. Terminology

The anatomical terms follow Davies [34], Bode [29], Snodgrass [35], Snodgrass [36], and Zhou and Rédei [37]. Detailed descriptions of the terminologies used are provided in Appendix A.

#### 2.3.2. Dissections

Female thrips specimens were dissected in a 0.1 M cacodylate buffer using #2 insect pins under an Olympus SZX16 stereomicroscope equipped with an Olympus SDF PLAPO 2X PFC objective (230× magnification). As the first step, we detached 3–4 apical segments from the rest of the abdomen. For CLSM, we separated the ovipositor assembly (ovipositor, T8 and T9).

#### 2.3.3. CLSM

Female ovipositors were fixed in 2.5% glutaraldehyde in a 0.1 M cacodylate buffer for 24 h at 4 °C. Specimens were washed in a 0.1 M cacodylate buffer (3 × 15 min), transferred into anhydrous glycerol and kept on concave slides for further examination. Specimens were imaged between two #1.5 coverslips with an Olympus FV10i confocal laser-scanning microscope (CLSM) at the Microscopy and Cytometry Facility at the Huck Institute of Life Sciences, located at the Pennsylvania State University. We used three excitation wavelengths (405 nm, 473 nm and 559 nm) and detected the autofluorescence using three channels with emission ranges of 400–480 nm, 490–590 nm and 570–670 nm. The resulting image sets (oib files) were assigned pseudo-colors that reflected the fluorescence spectra (400–480 nm: blue; 490–590 nm: green; 570–670: red). Volume-rendered micrographs and media files were created using FIJI [38,39].

#### 2.3.4. SBFSEM

Three or four apical segments of the abdomen were fixed in 2.5% glutaraldehyde and 6% sucrose in a 0.1 M cacodylate buffer for two hours at room temperature, then for 24 h at 4 °C. Following the aldehyde treatment, specimens were washed in a 0.1 M cacodylate buffer (3 × 15 min) and further processed following a modification of the SBFSEM protocol of Deerinck et al. [40]. Specimens were placed in 3% potassium ferrocyanide in a 0.3 M cacodylate buffer in osmium tetroxide (EMS) for 1 h. Tissues were washed (3 × 15 min) with nanopure water and placed in a TCH solution for 20 min. After that, the tissues were washed in water (3 × 15 min) and incubated in 1% uranyl acetate at 4 °C for 24 h. Next, the tissues were incubated in an aspartic acid solution (Walton lead aspartate staining) followed by dehydration in 25%, 50%, 75%, 85%, 95% and 100% ethanol solutions, and were then fixed in Eponate (Ted Pella incorporation, Redding, CA, USA). The molds, prepared with 100 % resin, were placed in the oven at 70 °C for 48 h. Specimen-containing resin blocks were trimmed and sectioned with a Leica UCT ultramicrotome, Germany, placed on an aluminum pin with conductive silver glue and sputter coated with gold/palladium. Blocks were sectioned and imaged using a Zeiss SIGMA VP-FESEM (Augsburg, Germany) with a Gatan 3View2 accessory at an accelerating voltage of 3.0 kV with a vacuum level of 40 Pa. The voxel size was 0.08 µm × 0.08 µm × 0.2 nm and the total image size was 7910 × 5516 pixels. AMIRA software 5.5.0 Ink was used for alignment, segmentation and image processing.

## 3. Results

The morphology of female thrips is composed of two panoistic ovaries, oviducts attached to each ovary, spermatheca, accessory glands, the appendage gland, a common oviduct and a vagina.

### 3.1. Female Terminalia of N. variabilis

#### 3.1.1. Ovaries

Two ovoid ovaries are present, with each composed of eight ovarioles (Figure 1 and Figure 2). One ovary measures 36.33 µm wide and 73.66 µm long, and one ovariole measures 8.86 µm wide and 57.93 µm long. The inter-ovular space is filled with thick masses of cells and fluid. The ovary is surrounded by soft tissues (Figure 3). The maturing oocytes are present proximally and the newly growing oocytes are present distally near the germarium (Figure 4). The oocytes are lined with epithelium (ET) (Figure 3). The right ovary in our dissection contained 17 oocytes, which were lined together and surrounded by follicle cells (FC) (Figure 3).

#### 3.1.2. Oviducts

Two oviducts arise from the ovaries, joining distally near the genital chamber (Figure 2) (Appendix A). Each oviduct is approximately 48.19 µm long and is connected distally with the oviductus communis. The oviducts are lined with a single-layer epithelium (Figure 2).

#### 3.1.3. Spermatheca

The spermatheca is 25.73 µm long and 19.05 µm wide (Figure 4), and is connected to the ovipositor by three muscles. The spermatheca is dorsal to the oviductus communis, and stores the sperm cells (Figure 4). The spermatheca is filled with large, dense cells of uncertain function and origin. The spermatheca has a very complicated pathway or structure inside, which is either muscular or has a thick mass of cells.

#### 3.1.4. Accessory Glands

Accessory glands are present on the dorsal side of the spermatheca, and open in the oviductus communis. The accessory glands are about 39.59 µm long and 8.50 µm wide (Figure 5A) (Appendix A). The two accessory glands are present on the left side of the oviducts and above the spermatheca (Figure 5B). These glands open into the vagina. The accessory glands are composed of different types of cells: skin cells (sc), basement membrane cells (mb), gland cells (gc) and wall cells (wc) (Figure 5A).

#### 3.1.5. The Appendage Gland

The appendage (ADR) gland is large and lies ventral to the right ovary and dorsal to the ovipositor. The ADR gland is trapezoidal in shape (Figure 6A) (Appendix A) and measures 37.30 µm wide and 74.44 µm long. The two oviducts are located laterally to the appendage gland (Figure 2 and Figure 7) The ADR gland is composed of Type III gland cells: Drusen cells (DZ) and channel cells (KZ) (Figure 6A). The ADR gland is connected through the ADR gland duct to the ADR orifice (Figure 6B). The ADR gland duct is 26.89 µm long and 4.2 µm wide (Figure 6B), and the gland opening is circular with a diameter of 3.62 µm (Figure 6B).

#### 3.1.6. Common Oviduct and Vagina

The two short lateral oviducts join to form the common oviduct. The spermatheca (sth, Figure 7) is present at the front end of the vagina (vag, Figure 7) and is connected to the wall of tergum VII with the branched tergo spermatheca muscles (tsm1, tsm2 and atsmps) (Figure 4 and Figure 7) (Appendix A). The vagina opens into the genital chamber (gch, Figure 7). The vagina is connected to the ovipositor via muscles viz., the spermatheca vagina muscle (svm) (Figure 7). The spermatheca basivalvulae muscle (sbm) originates from the spermatheca and is inserted in the ovipositor at the basivalvulae.

### 3.2. Muscles of the Internal Genitalia

The internal genitalia are lined with muscles that are attached to the ovaries, appendage gland, spermatheca, oviducts, accessory glands and the oviductus communis (Appendix A). The spermatheca is connected to the branched tergo spermatheca muscle (tsm), which consists of three fibers viz., tergo spermatheca muscle 1 (tsm1), tergo spermatheca muscle 2 (tsm2) and antero tergo spermatheca muscle postero spermatheca (atsmps) (Figure 7) (Appendix A). Two muscles, tsm1 and tsm2, are attached to the body wall and the dorsal side of the apodeme (Figure 7). These two muscles are thick, with a diameter of 10.75 µm and 9.64 µm. A third muscle, the antero-tergo spermatheca postero spermatheca muscle (atsmps), is directly connected to the spermatheca. This muscle is 17.26 µm long and is connected laterally to the spermatheca as well as to the other two muscles (tsm1 and tsm2) (Figure 7). A fourth muscle, the spermatheca basivalvulae muscle (sbm), is connected dorsally to the spermatheca and ventrally to the basivalvulae (BV, Figure 7). This muscle is 16.77 µm long. The appendage gland, i.e., the latero-tergo-postero appendage gland (ltpag), is connected to five muscles. This muscle is a branched muscle and consists of five fibers viz., latero-tergo-postero appendage gland branches 1–5 (ltpagmb1–5) (Figure 7). Two of these muscles are attached to the ovary as well. The latero-tergo-postero appendage gland muscle branch 1 (ltpagmb1) (colored blue in Figure 7) is 9.83 µm thick and 23.73 µm long and inserts dorsally into the ADR gland. The latero-tergo-postero appendage gland muscle branch 2 (lptagmb2) is relatively small, being 8.71 µm thick and 9.72 µm long (Figure 7), and joins the latero-tergo-postero appendage gland muscle branch 4 (lptagm4) near its attachment to the ADR gland (Figure 7). The latero-tergo-postero appendage gland muscle branch 4 (lptagmb4) is 11.29 µm thick and 25.04 µm long, and attaches to the ovary laterally (Figure 7). However, it seems that the ovary receives most of its support directly from the ADR gland attachment because the ADR gland is very close to the ovary. The latero-tergo-postero appendage gland muscle branch 3 (lptagmb3) is relatively short, being 11.35 µm thick and 15.69 µm long. The latero-tergo-postero appendage gland muscle branch 5 (lptagmb5) is 14.76 µm long and 18.31µm wide, and connects the ovary and the ADR gland (Figure 7). Two muscles, viz., the muscle of the ovary ovipositor appendage gland (mooag) and latero-tergo-postero appendage gland muscle branch 5 (lptagmb5), support the ovary and the ADR gland (Figure 7). These muscles are also connected to the spermatheca on one side. Through these two muscles, the ADR gland, ovary and spermatheca are connected to one another. The spermatheca vagina muscle (svm) is connected to the spermatheca. The ovary ovipositor appendage gland muscle (mooag) connects the ovary and the ADR gland.

### 3.3. External Genitalia

Abdominal Segment VIII consists of the ovipositor, valvifer (vf), basivalvulae (BV) and valvulae (Figure 8 and Figure 9) (Appendix A). Segment IX consists of the second ventral valvulae. Segment XI is a very small segment and bears the epiproct and paraproct (Ep and pp in Figure 8A).

Segment VIII is a comparatively larger sclerite than Segments IX and X. The ovipositor shaft has a sharp pointed tip (Figure 8B and Figure 9) (Appendix A). When the ovipositor is at rest, it is encircled dorsally by the genital membrane (gm: Figure 8). The first valvifer is dorso-laterally elongated and rod shaped, and is joined to the wall of the genital chamber on their ventral side. The lateral sides of the first valvifer are attached to Basivalvulae 1 and 2. At the other end, the first valvifer is attached to the apodeme, which is attached through ovipositor muscle 1 (ovm1) to the abdomen. The first valvifer is connected to the first valvula anteriorly (or proximally; 1vv, 1vf: Figure 8 and Figure 9).

#### 3.3.1. Genital Chamber

The genital chamber dorsally surrounds the ovipositor base, valvifer, first valvula, second valvula and ADR gland opening (Figure 8A and Figure 9).

#### 3.3.2. Basivalvulae (BV 1 and 2)

BV 1 and 2 are small sclerites that are connected to the base of the first valvifers (Figure 9 and Figure 10). BV1 is 2.73 μm long, and BV2 is 1.22 μm long (Figure 9). The basivalvulae are attached to the valvulae and the ventral ramus; however, their inclination is downward. A muscle, the spermatheca basivalvulae muscle (sbm, Figure 7), is connected to the basivalvulae and the spermatheca.

The ovipositor consists of valvifers 1 and 2 (Vf1 and Vf2), valvulae 1 and 2 (V1 and V2), spines, the basivalvule (BV1 and BV2), the dorsal ramus and the ventral ramus (VR1 and VR2). The basivalvulae are triangular (Figure 10). Valvifer 2 is swollen compared with Valvifer 1. The first valvifer is 7.53 μm long and is connected to the apodeme through an articulation/joint (1vf; Figure 10 and Figure 11). The first valvifer is dorso-laterally elongated in the dorsal view (Figure 9 and Figure 10) with a straight anterior margin (Figure 10 and Figure 11). This intervalvifer articulation is present at the dorsolateral point between the first valvifer and the apodeme (Figure 10 and Figure 11). The 1Vf is connected to the first valvulae by ventral ramus 1.

The first valvula (1.V1) is elongated. The 1.V1 runs along the length of the ovipositor and is about 47.22 µm long (Figure 8, Figure 9 and Figure 10). The first valvula is sclerotized in some places: (1) at the junction with the dorsal ramus 1 and (2) in certain areas with the ventral ramus. The spines and teeth are present on the ventral ramus and first valvula (Figure 9) (Appendix A). The second valvifer is 7.28 µm long (Figure 8, Figure 9 and Figure 10). It is similar to the first valvifer but is slightly distended and triangular. The second valvifer (2.vf) is slightly distended/swollen as a depression in apodeme IX. The second valvifer (2.vf) is attached to the second valvulae (2.VI). The second valvulae are attached to the second valvifers and continue along with the first valvula (Figure 8, Figure 9 and Figure 10). The second valvulae are 43.74 µm long. The second valvulae are narrowed at the tip and form the egg canal together with the first valvulae. The second valvulae are barbed. The size of the tip of the ovipositor shaft is 3.35 µm (Figure 8) (Appendix A).

#### 3.3.3. Ventral and Dorsal Ramus

The ventral ramus is attached to the first valvulae, and the dorsal ramus is attached to the second valvulae (Figure 10).

### 3.4. Muscles of the External Genitalia (Segments VIII, IX and X)

Four muscles, viz., tergo valvilis posterior muscle 9 (mtvp9), tergo valvilis primus muscle 8 (mtvp8), tergo valvilis medial muscle 10 (mtvm10) and longitudinal inter tergite muscle 10 (mlit 10) are attached to the ovipositor. Among these, tergo valvilis posterior muscle 9 (mtvp9) is composed of five branched muscles, viz., ovm1–5 (Figure 11 and Figure 12). Ovipositor muscle 1 (ovm1) originates from tergum and is inserted into the second valvifer. This muscle fiber is laterally attached to the apodeme (Figure 11).

Distally, this muscle is connected to ovipositor muscle 2 (ovm2) and ovipositor muscle 5 (ovm5). Ovipositor muscle 2 (ovm2) arises from the tergum and is inserted into the second valvifer. This muscle is connected laterally to another strong muscle, ovipositor muscle 3 (ovm3) (Figure 11). Ovipositor muscle 3 (ovm3) originates from the tergum and is inserted into ovipositor muscle 2 (ovm2) and ovipositor muscle 4 (ovm4). On its anterior side, ovipositor muscle 2 (ovm2) is connected to ovipositor muscle 5 (ovm5) (Figure 11 and Figure 12). Though ovipositor muscles 2 and 5 can appear to be one muscle, the two muscles are separate, as shown in the SEM image (Figure 11 and Figure 12). Ovipositor muscle 4 is connected posteriorly to the tergum and is inserted laterally with ovipositor muscle 3 (ovm3) and the valvulae (Figure 11 and Figure 12). Ovipositor muscle 5 (ovm5) is connected dorsally to ovipositor muscles 3 and 4 to the ovipositor shaft (Figure 11 and Figure 12). Tergo valvilis primus muscle 8 (mtvp8) originates from the tergum and it is inserted into apodeme plate 8. It is connected to the body wall and the ovipositor at tergum 10 (Figure 11). Tergo valvilis medial muscle 10 (mtvm10) is an unbranched muscle which originates from the tergum and is inserted into the second valvula’s tip. Longitudinal inter tergite muscle 10 (mlit10) is an unbranched muscle which originates from transverse tergite 11 and is connected to tergite 9 (Figure 11).

### 3.5. Morphology of Mature and Immature Eggs

Mature eggs were kidney-shaped and the immature eggs were oblong (Figure 13).

## 4. Discussion

We examined the structure of the female reproductive system of *N. variabilis* to provide a detailed view of the mechanism of egg movement inside the insect and the method of egg-laying into the leaf sheath. We documented the morphology and musculature of the individual structures of the female reproductive system. This is the first study describing the detailed structure of the ovary, ovarioles and the ovariole arrangement in the subfamily Sericothripinae.

We found paired ovaries, in which the ovarioles are arranged. The inter-ovular space is filled with thick masses of fluid or cells. Beneath the ovaries, the ADR gland is present. The ovaries are connected to the oviducts. Two muscles, viz., the muscle of the ovary oviductus appendage gland (mooag, Figure 7) and latero-tergo-postero appendage gland branch 5 (ltpagmb5) connect the ovary to the ADR gland and spermatheca. Five muscles (ltpagmb1, ltpagmb2, ltpagmb3, ltpagmb4 and ltpagmb5) connect the ADR gland to the body wall. Among these five muscles, one muscle (ltpagmb5) connects the ADR gland to the ovary. Three muscles (tergo spermatheca muscle 1, tergo spermatheca muscle 2 and antero-tergo spermatheca muscle postero spermatheca (tsm1, tsm2, atsmps)) connect the spermatheca to the body wall; one muscle (mooag) connects the spermatheca to the ovary and the ADR gland; one muscle (the spermatheca vagina muscle (msv)) connects the spermatheca to the vagina and one muscle (the spermatheca basivalvula muscle (sbm)_ connects the spermatheca to the basivalvulae (Figure 7). The external genitalia are lined with seven muscles which regulate the movement of the ovipositor.

Although work on the internal and external genitalia of different thrips species has been conducted several times, a detailed description of the structures and egg-laying mechanism was provided by Bode [29], while Heming [20] provided a detailed description of the evolution of arrhenotoky with reference to germline cells and chromosomes. Bode [29] described the structures of the ovipositor, spermatheca and appendage gland; the musculature of the female reproductive system; and the egg-laying mechanism in *Thrips physapus*. We found similarities to and differences from Bode [29]. Bode [29] described the appendage gland as the accessory gland. However, our detailed SBFSEM images and ultrastructure analysis showed that the appendage gland is not an accessory gland but an additional gland attached to the ovipositor that opens at the ADR gland’s opening through the ADR gland duct [29]. Thrips instead have an accessory gland and an additional ADR gland (Figure 6 and Figure 7).

Moreover, the basivalvulae in *Thrips physapus* were similar to those of *N. variabilis* except that the BV is not connected to muscle 8; instead, it is connected to the spermatheca basivalvulae muscle (sbm), which connects to the spermatheca. Bode [29] explained that muscle 8 is connected to the body wall; instead, in *N. variabilis*, the spermatheca basivalvulae muscle (sbm) is connected on its ventral side to the spermatheca and on its dorsal side to the BV1. We hypothesize that when the egg reaches the vagina, the size of the egg creates pressure which opens the valve of the spermatheca. The sperm then moves through the pathway and enters the vagina to fertilize the egg. Due to pressure, the muscle bends down, the basivalvulae stretch and the egg enters the ovipositor.

Bode [29] described the teeth and spines in the ovipositor on the ventral ramus and first valvulae. We also observed the teeth and spines in the ventral ramus and the spines on its outer side. However, the pattern of the spines and teeth in Bode’s [29] diagram was not similar to ours. Instead, we observed that the teeth and spines are at the same height and one spine is not equal to two teeth, as mentioned in Bode’s illustration. We assume that the teeth work as a valvillus, a structure found in the egg canal of *Homolobus truncator* (Hymenoptera: Ichneumonoidea: Braconidae) [41]. We suppose that hydrostatic pressure and the teeth play a role in the movement of eggs in the egg canal, but after the egg reaches the ADR gland’s opening, the gland secretions push the egg into the host leaf. Bode [29] described M13 in cross-section but did not explain how it connected to the spermatheca and basivalvulae. The position of M13 corresponds to the spermatheca basivalvulae muscle (sbm), but we described, for the first time, how this muscle connects to the basivalvulae. Bode [29] described how M17 attaches the spermatheca to the ovary. However, we found three muscles (tsm1, tsm2, atsmps) which attach the spermatheca to the body wall, and one muscle (msv) which connects the spermatheca to the vagina. One muscle (mooag) provides support to the appendage gland but it is also close to the ovary at its base (Figure 7). Although Bode [29] explained that the spermatheca is connected to the vagina, ovary and ovipositor through muscle 2. Muscle 2 in Bode [29] is the muscle of the ovipositor oviductus ADR gland (mooag) in our study.

Bode [29] did not illustrate the ADR gland’s connection through the spermatheca basivalvulae muscle (sbm) to the spermatheca. Our SBFSEM results showed that the ADR gland is a big gland and is supported by the latero-tergo-postero appendage gland muscle to the body wall and ovary. This latero-tergo-postero appendage gland muscle and ovary muscle (mooag) were not described in Bode [29] previously. Moreover, Bode [29] did not provide a detailed illustration of the ADR gland connecting the ADR gland’s duct and the ADR gland’s opening, and its muscles’ orientation with respect to the oviduct and the ovary in detail. This is the first illustration of the musculature of the interconnection of the ADR gland, ovaries and spermatheca in female thrips.

Although Bode [29] described the egg-laying method of the female thrips, our SBFSEM 3D AMIRA model has been able to provide more insight into the mechanism of egg-laying in *N. variabilis*. The mature egg moves from the ovaries into the lateral oviducts, which join to make the vagina. The ovary is joined to a muscle (mooag). The movement of the egg through the oviducts and oviducts communis may be due to the adhesion and contraction of the lateral oviduct and oviductus communis; however, the muscle (mooag) and its role may be important in the movement of eggs inside the oviducts.

The opening of the spermatheca is immediately in front of the oviductus communis. When the egg reaches the spermatheca, sperm moves from the spermatheca to the vagina to fertilize the eggs. The fertilized eggs move into the ovipositor. The vagina is held by the spermatheca vagina muscle (svm) and connected to the ovipositor through a duct. An egg moves through the egg canal by the movement of the first and second gonapophysis, and simultaneous contraction and relaxation of the vagina. When the egg reaches the proximal end of the ovipositor, the basivalvulae holds the egg and stretching is induced. The movement of the gonapophysis, which is regulated by the valvifer attached to the apodeme and tergo valvilis posterior muscle 9 helps in the movement to the ADR gland’s opening. Bode [29] explained that when the egg reaches the ADR gland, the secretion from the ADR gland pushes the egg down. Our work supports the findings of Bode [29].

The ovipositor moves out of tergite 8 and is held perpendicular to the leaf surface. The leaf is punctured by the first and second valvulae. The valvula continues penetrating until the whole ovipositor is present in the leaf. The pointed tip helps to cut the leaf tissue and position the ovipositor. Through the movement of the first and second valvulae and the muscular movement of tergo valvilis muscle 9, tergo valvilis medial muscle 10 and longitudinal intertergite muscle 10, the egg emerges from the ovipositor slit, probably with the aid of the secretions from the ADR gland. The secretions of the ADR gland act as a lubricant for the movement of the egg through the ovipositor. Once the egg is laid, the ovipositor retracts. The retraction of the ovipositor may be regulated by different processes, such as membrane sheath elasticity and muscles, viz., tergo valvilis medial muscle 10 (mtvm10) and longitudinal intertergite muscle 10 (mlit10).

The structures found in the ovipositor of *N. variabilis* were similar to those described by Melis [27,42], except that the third valvulae or gonoplac was absent. However, Scudder [43] described the ovipositor in *T. validus* as consisting of the first valvula, first valvifer, the gonangulum, second valvifer, and no gonoplac. The gonangulum in Scudder [43] is similar to what we observed; however, the basivalvulae in *N. variabilis* is not rod-shaped but triangular, compared with *Thrips validus*.

The ovipositor in Hemiptera consists of the first and second valvifers (vf), which are dissimilar in size and often fused with other tergites in the abdomen (e.g., Pyrrhocoridae) Scudder [44]. In Thysanoptera, the first and second valvifers (vf) are not fused with other tergites, though they are attached to the apodeme [42]. The first and second valvifers (vf) in Hemiptera (Cicadellidae) are dissimilar in shape but the second valvifer (vf) is attached to the second valvula (V2) and gonangulum (Go), and is particularly visible, not fused with any other tergite or sclerite [43]. The first and second valvulae (V1 and V2) in Thysanoptera are elongated and have barbs or lacinations, which are similar to those of Hemiptera, in which the first and second valvulae (V1 and V2) are lacinated. As in Hemiptera (Tingidae), we also noticed the first and second ramus, which form interlocking ridges.

The laciniated gonapophysis of Terebrantia provides the ecological advantage of occupy the internal leaf niches, which the other suborder, Tubulifera, cannot.

In Hymenoptera, there is a venom gland which can serve to propel or push the eggs into the host [41]. In Ceraphronoidea, the second valvifer (2. vf) is also swollen or distended near the anterior apex [45]. The ridges/spines were also found on the apical portion of the valvulae in dragonflies [46]. We also found similar ridges in thrips, which were also similar to those found by Bode [29] in *Thrips physapus*. Some authors proposed the role of the teeth present on the apical valvulae as stops, which prevent the lower valve from being extended posteriorly compared with the upper valve [47].

## 5. Conclusions

Overall, during the morphological studies of the internal and external genitalia, we found that the internal genitalia in female *N. variabilis* are composed of two ovaries, oviducts, accessory glands, the spermatheca, the ADR gland, the common oviduct and the vagina. The external genitalia consist of the apodeme, valvifers, valvulae, and the dorsal and ventral ramus. The ovary, spermatheca, oviducts, ADR gland, vagina and external genitalia are connected through several muscles which enable the transfer of mature eggs from the ovaries to the parenchymatous sheath of the leaf. Female *N. variabilis* lay eggs near the mid-rib or veins in the parenchymatous leaf tissues, but not in the inter-venal area, unlike other species of Thripidae. Similar to *Thrips physapus*, female *N. variabilis* also have a large ADR gland.

## Figures and Tables

**Figure 1 insects-13-00566-f001:**
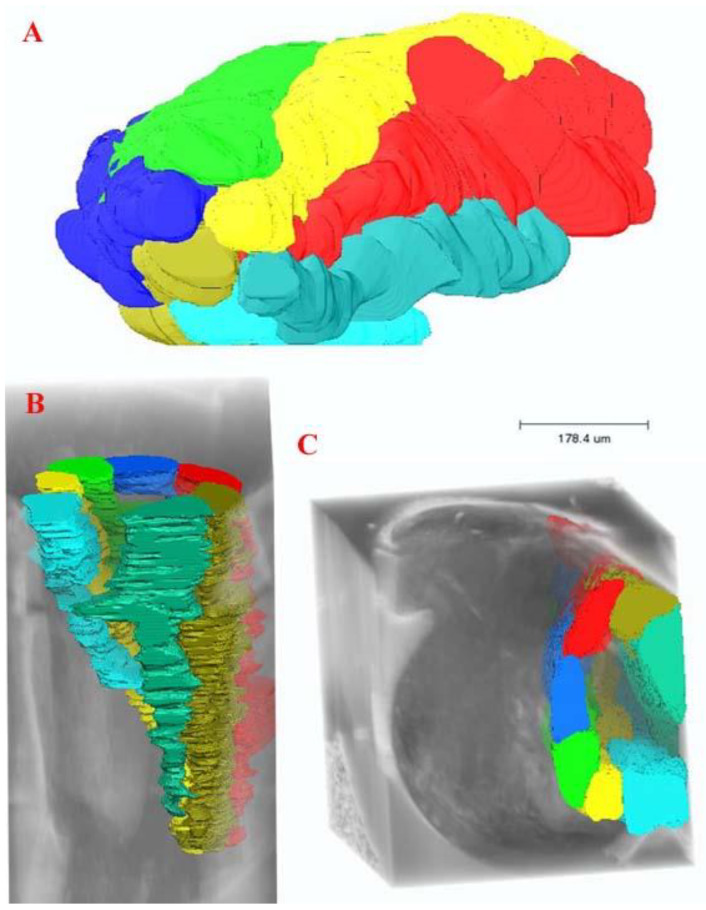
Arrangement of the ovarioles in the ovary. (**A**) Ovariole arrangement without the background of the whole female reproductive system. (**B**) Anterior view of the arrangement of the ovarioles inside the ovary. (**C**) Lateral view of the ovarioles inside the ovary.

**Figure 2 insects-13-00566-f002:**
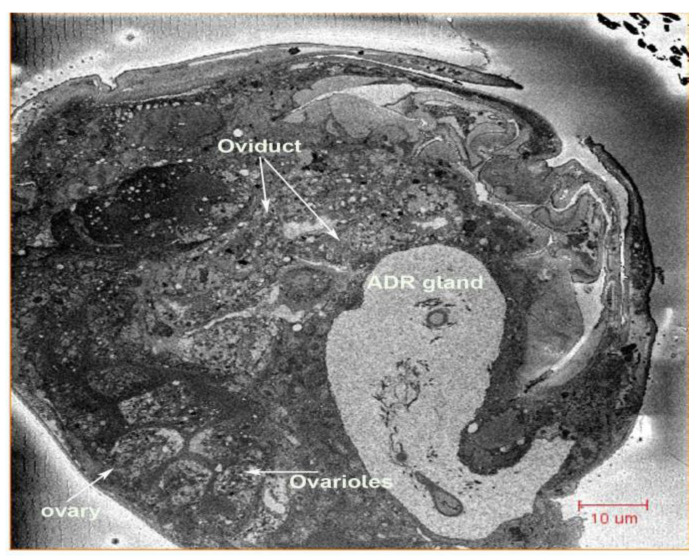
Cross-section of the female, showing the appendage (ADR) gland, ovary, oviducts and genital chamber.

**Figure 3 insects-13-00566-f003:**
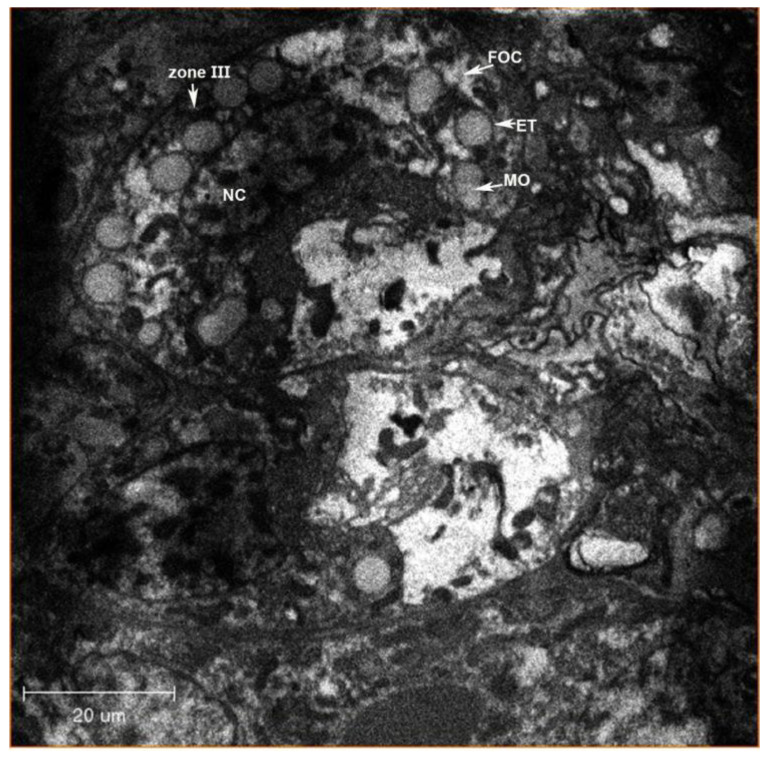
Longitudinal cross-section of the ovariole, showing oocytes in *N. variabilis*. The various parts of the ovariole consist of oocytes (immature and mature), yolk cells, follicle cells and nutrition cells. Abbreviations: ET, epithelial layer; MO, mature oocytes; IO, immature oocytes.

**Figure 4 insects-13-00566-f004:**
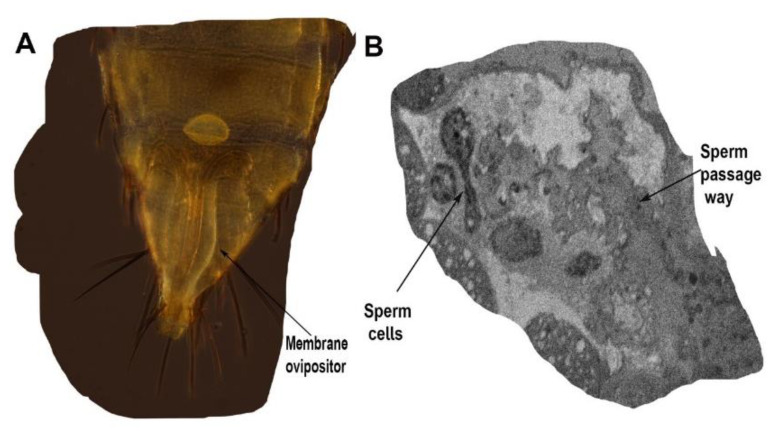
Spermatheca. (**A**) Ventral view of female lower abdomen showing the spermatheca (S) and ovipositor lined with the ovipositor membrane. (**B**) Spermatheca cross-section showing the sperm cells and passageway that connects to the vagina.

**Figure 5 insects-13-00566-f005:**
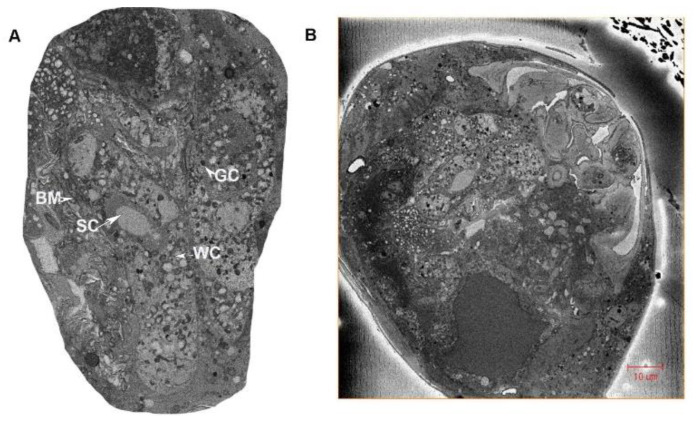
Cross-section of female terminalia showing the accessory gland cells in the female thrips. (**A**) Detailed view of accessory gland cells consisting of gland cells (gc), skin cells (sc), wall cells (wc) and membrane cells (bm). (**B**) Detailed overview showing the connection of the accessory gland to the ovipositor.

**Figure 6 insects-13-00566-f006:**
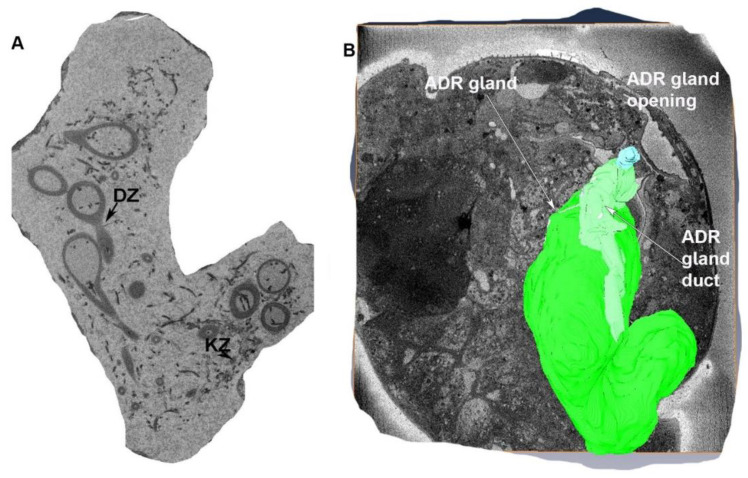
(**A**) Cross-section of the ADR gland showing the glandular cells (DZ) and channel cells (KZ). (**B**) Connection of the ADR gland to the ADR gland duct, ovary and ovipositor. Here ADR gland is represented in green color while ADR gland duct is represented in light green color.

**Figure 7 insects-13-00566-f007:**
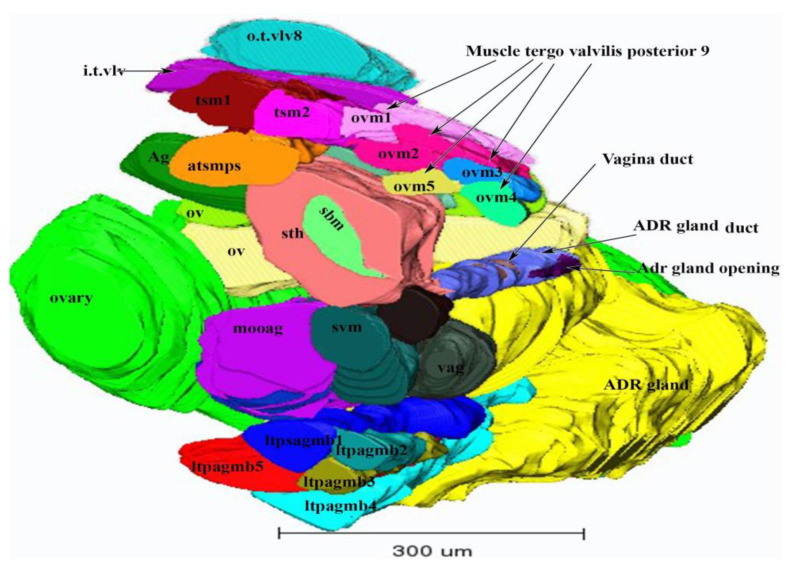
A 3D model depicting the female reproductive system of *N. variabilis*, showing the internal genitalia and muscles. Abbreviations: ltpagmb 1–5, latero-tergo-postero appendage gland branch 1–5; mooag, muscle of the ovipositor oviduct accessory gland; ADR, appendage gland; sth, spermatheca, sbm, spermatheca basivalvulae muscle; svm, spermatheca vagina muscle; vag, vagina; tsm, tergo spermatheca muscle 1; tsm2, tergo spermatheca muscle 2; atsmps, antero-tergo spermatheca muscle postero spermatheca. The tergo valvilis posterior muscle 9 consists of five branched muscles: ovm1–5, i.t.vlv (inter-tergal valvilis muscle), o.t.vlv 8 (outer tergo valvilis muscle 8) and ag (accessory gland).

**Figure 8 insects-13-00566-f008:**
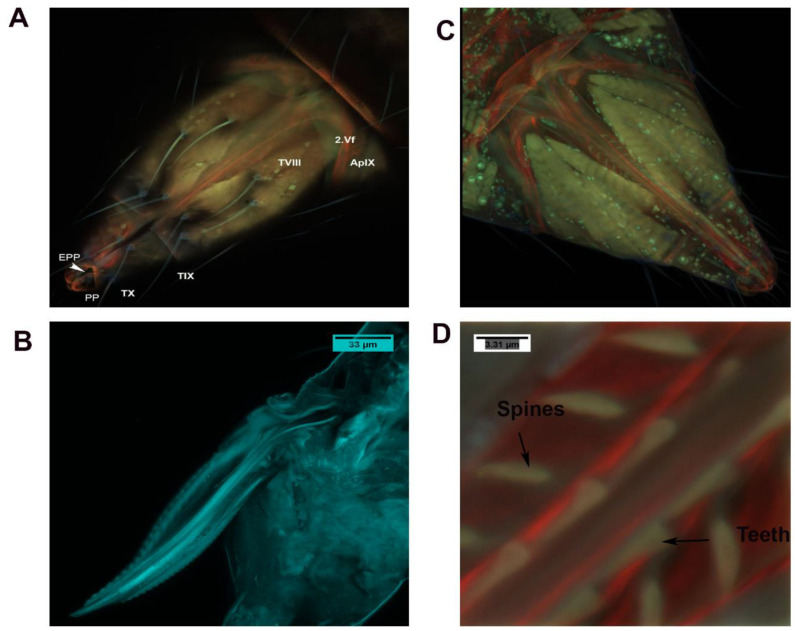
Ovipositor and genital chamber. (**A**) Ventral overview of the female ovipositor showing the apodeme, the second valvifer and the genital chamber in the 8th, 9th and 10th sternum. (**B**) Lateroventral overview of the female ovipositor out of the female abdomen. (**C**) Ventral overview of the female ovipositor inside the membrane with the musculature. (**D**) Ventral overview of the female ovipositor showing the spines and teeth in the ovipositor to aid in the movement of eggs inside the egg canal.

**Figure 9 insects-13-00566-f009:**
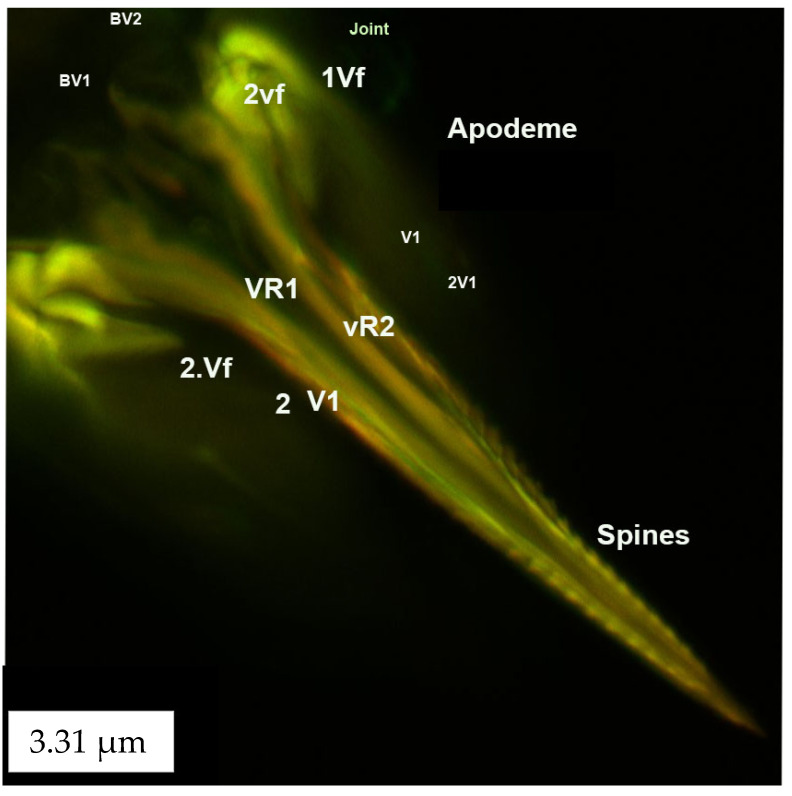
CLSM volume-rendered micrograph showing the ventral view of the ovipositor of *Neohydatothrips variabilis*.

**Figure 10 insects-13-00566-f010:**
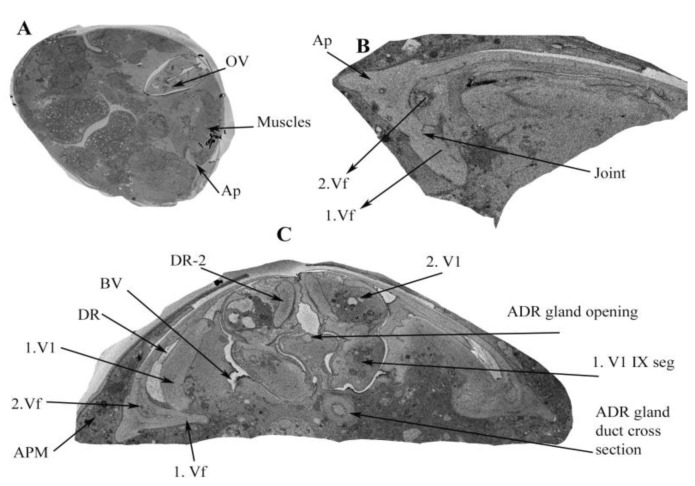
SBFSEM micrograph (horizontal cross-section) showing the various parts of the thrip’s ovipositor. (**A**) Apodeme (AP), IXth segment and muscles supporting the ovipositor and apodeme. (**B**) Apodeme, second valvifer and first valvifer explained. (**C**) Apodeme (Ap), second valvifer (2.Vf), first valvifer (1.Vf), first valvulae (1.V1), dorsal ramus (DR), ventral ramus (VR), and ninth segment valvulae V1 and V2 explained.

**Figure 11 insects-13-00566-f011:**
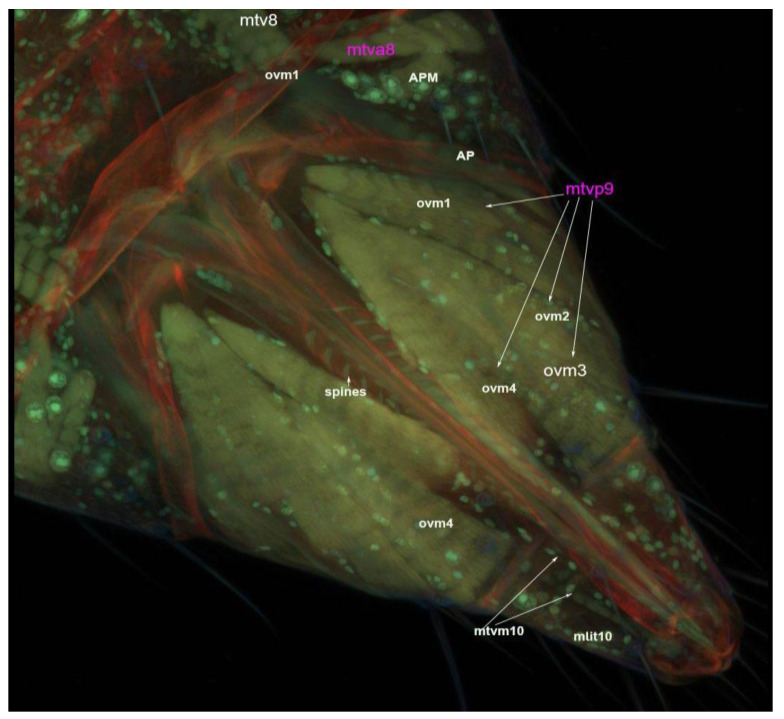
CLSM image of the ventral side of the female thrip’s ovipositor along with its muscles. ovm1–5, ovipositor muscles 1–5; mtvm10, tergo valvilis medial muscle 10; mlit10, longitudinal intertergite muscle 10; mtvp9, tergo valvilis primus muscle 9; mtv8, tergo valvilis primus muscle 8; mtv8a, tergo valvilis anterior muscle 8; APM, apodeme muscle.

**Figure 12 insects-13-00566-f012:**
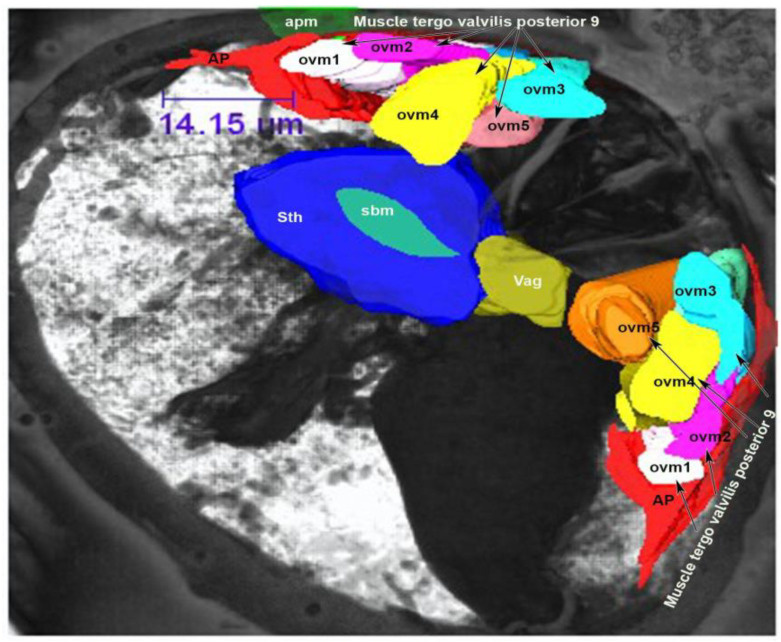
AMIRA 3D model depicting the external genitalia muscles. ovm1–5, ovipositor muscle 1–5; vag, vagina; sth, spermatheca; Ap, apodeme; Apm, apodeme muscle.

**Figure 13 insects-13-00566-f013:**
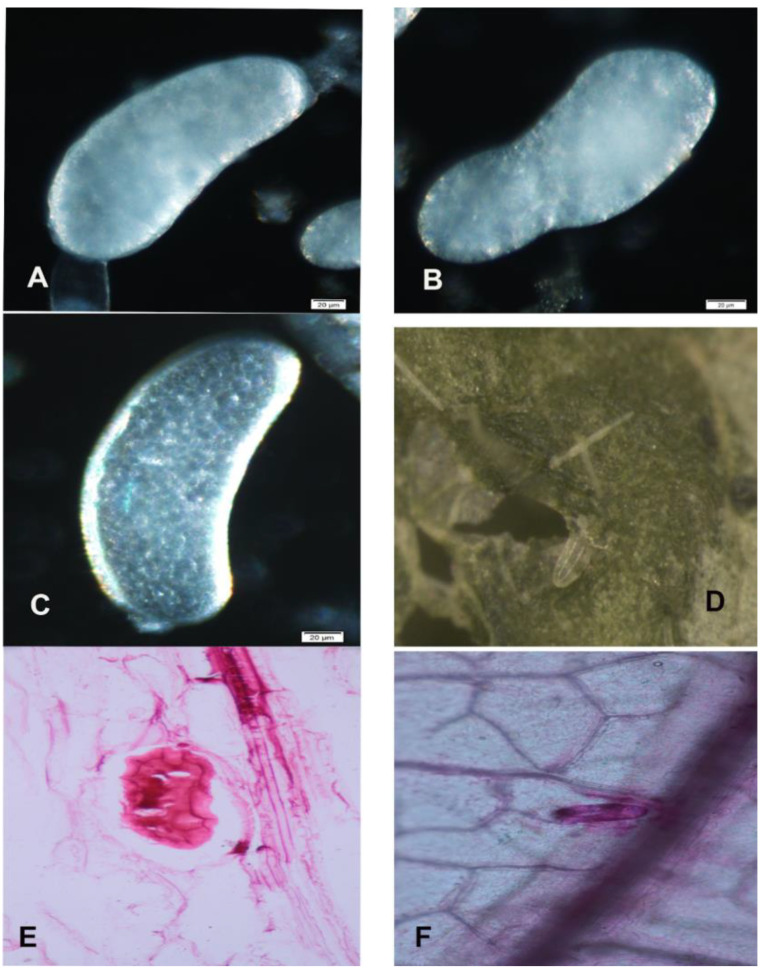
Light microscopic view of thrips eggs dissected and inside the leaf sheath. (**A**,**C**) Mature egg dissected from the ovary. (**B**) Immature egg. (**D**) Egg present inside the leaf. (**E**) Cross-section of a leaf, showing the stained egg. (**F**) Stained egg inside the leaf near the mid-rib.

## Data Availability

Appendix A is available in the Appendix A.

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
