# Peer review of "Morphology of the Female Reproductive System of the Soybean Thrips, Neohydatothrips variabilis (Beach, 1896) (Thysanoptera: Thripidae)"

_insects, 2022, doi:10.3390/insects13070566_

Round 1
Reviewer 1 Report
This is an interesting fundamental paper in which the authors report about morphology of female reproductive system of N. variabilis, an important vector of SVNV in North America and Egypt. After reading the article carefully, I find it suitable for publication, but only after minor revision. Before final acceptation of the paper for publication, I suggest to correct it in the following places:
Title
line 3: replace "Thrips" with "thrips"
line 4: delete ": Sericothripinae"
p. 26: replace "Soybean" with "soybean"
p. 46: I strongly suggest to add some sentences about thrips as crop pests and some suitable references, for example: STOPAR et al. 2021. Thrips and natural enemies through text data mining and visualization. Plant Protection Science, 57, 1: 47-58
p. 46: replace "variabilis) (Beach, 1896)" with "variabilis [Beach, 1896])
p. 61: replace "(Frankliniella occidentalis)" with "(Frankliniella occidentalis [Pergande])
p. 62: replace "(Thrips tabaci)" with "(Thrips tabaci Lindeman).
p. 86: replace "(L.)" with "L."
p. 99: add the latitude, altitude and height above the sea level for the location
p. 490: delete "L." after "Thrips physapus"
p. 501: "N. variabilis" should be written in italic
p. 502-503: "Thrips physapus" and "N. variabilis" should be written in italic
Author Response
Response to Reviewer 1 Comments
Comments and suggestion: This is an interesting fundamental paper in which the authors report about morphology of female reproductive system of N. variabilis, an important vector of SVNV in North America and Egypt. After reading the article carefully, I find it suitable for publication, but only after minor revision. Before final acceptation of the paper for publication, I suggest to correct it in the following places:
We appreciated the insightful and constructive comments made by the reviewers. We have made numerous revisions in response to these comments and we feel that these changes have considerably strengthened the paper.
A point by point response to the reviewers comments is presented below, with our comments in bold text.
Point 1: Title
line 3: replace "Thrips" with "thrips"
Response 1: We have replaced the word “Thrips” was replaced with “thrips”.
Point 2: line 4: delete ": Sericothripinae"
Response 2: We have deleted Sericothripinae.
Point 3:p. 26: replace "Soybean" with "soybean"
Response 3: We have replaced “Soybean” with “soybean”
Point 4: p. 46: I strongly suggest to add some sentences about thrips as crop pests and some suitable references, for example: STOPAR et al. 2021. Thrips and natural enemies through text data mining and visualization. Plant Protection Science, 57, 1: 47-58
Response 4: We have added this paragraph in the paper
Line 47-52
Thrips belonging to family Thripidae order Thysanoptera are economically important phytopagous and phytosanitary pests [1,2] because of the damage caused by oviposition, feeding and viruses transmission [3]. Among 290 genera and around 2400 species of Thripidae [4] a few species viz., Frankliniella occidentalis [Pergande], Thrips palmi [Karny], Thrips tabaci [Lindeman], Scirtothrips dorsalis[Hood] and Neohydatothrips variabilis [Beach] are important pests..
Point 5:p. 46: replace "variabilis) (Beach, 1896)" with "variabilis [Beach, 1896])
Response 5: We have replaced “variabilis (Beach 1896) with “(N. variabilis Beach, 1896)”
Point 6:p. 61: replace "(Frankliniella occidentalis)" with "(Frankliniella occidentalis [Pergande])
Response 6:
We have replaced “Frankliniella occidentalis” with “(Frankliniella occidentalis [Pergande])”
Point 7:p. 62: replace "(Thrips tabaci)" with "(Thrips tabaci Lindeman).
Response 7:
We have replaced “(Thrips tabaci)” with “(Thrips tabaci L.)”. As in line 50 the word Thrips tabaci [Lindeman] is present.
Point 8:p. 86: replace "(L.)" with "L."
Response 8: Replaced Thrips physapus (L.) with Thrips physapus L.
Point 9:p. 99: add the latitude, altitude and height above the sea level for the location
Response 9: Latitude, altitude, and the above sea level was added (40.71540, -77.94290 and 200 meters above the sea level)
Point 10:p. 490: delete "L." after "Thrips physapus"
Response 10: We have deleted “L.” after “Thrips physapus”
Point 11:p. 501: "N. variabilis" should be written in italic
Response 11: We have changed N. variabilis to N. variabilis
Point 12: p. 502-503: "Thrips physapus" and "N. variabilis" should be written in italic
Response 12: We changed the font from regular to Italic of Thrips physapus and N. variabilis
References:
- Wu, S.; Tang, L.; Fang, F.; Li, D.; Yuan, X.; Lei, Z.; Gao, Y. Screening, efficacy and mechanisms of microbial control agents against sucking pest insects as thrips. Advances in Insect Physiology 2018, 55, 199-217.
- Kashkouli, M.; Khajehali, J.; Poorjavad, N. Impact of entomopathogenic nematodes on Thrips tabaci Lindeman (Thysanoptera: Thripidae) life stages in the laboratory and under semi-field conditions. Journal of Biopesticides 2014, 7, 77.
- Stuart, R.R.; Gao, Y.-l.; LEI, Z.-r. Thrips: pests of concern to China and the United States. Agricultural Sciences in China 2011, 10, 867-892.
- Mound, L.A.; Morris, D.C. The insect order Thysanoptera: classification versus systematics. Zootaxa 2007, 1668, 395-411.

Reviewer 2 Report
Soy Bean thrips female reproductive system – Reviewer comments.
Line 5 – I suggest that this record of Neohydatothrips variabilis from “the Middle East” is probably a misidentification of Neohydatothrips abnormis, a similarly coloured species from the Mediterranean region. The authors provide no authentic reference for their identification – who identified it and based on what comparative information? The references quoted (1-4) are all by virologists who would not be competent to identify species in this complex worldwide genus of 120 described species. INSECTS will be doing biological science a great disservice if they agree to publish this unverified, and presumably incorrect, information.
Lines 55-65. This paragraph is irrelevant – UNLESS the author can produce evidence of thelytoky in their target species variabilis.
Lines 66-71 – again irrelevant since no information is provided on the sex ratio of variabilis.
Line 76 – it is not true that female Tubulifera do not have an ovipositor. I find it astonishing that no reference is made anywhere in this mss to the thrips morphological studies of Bruce Heming at Alberta.
Line 86 – Buckman et al (2013) did not recognise two clades of Thripidae – indeed we continue to have difficulty in recognising Thripinae as an independent clade.
I will leave my colleague who is a morphological specialist to comment on the main text, but I suggest that one or more of the co-authors for whom English is their native language should critically read the text – there are too many errors to be acceptable by most journals.
Laurence Mound 9.vi.2022
Author Response
Response to Reviewer 2 Comments
We appreciate the reviewer’s constructive comments and are glad that they also appreciated our work on morphology of female reproductive system of soybean thrips. We feel that these changes have improved the paper.
The paper passed through several changes, so the line numbers have been changed. I am writing the line numbers in the revised manuscript corresponding to the comments.
Point 1: Line 5 – I suggest that this record of Neohydatothrips variabilis from “the Middle East” is probably a misidentification of Neohydatothrips abnormis, a similarly coloured species from the Mediterranean region. The authors provide no authentic reference for their identification – who identified it and based on what comparative information? The references quoted (1-4) are all by virologists who would not be competent to identify species in this complex worldwide genus of 120 described species. INSECTS will be doing biological science a great disservice if they agree to publish this unverified, and presumably incorrect, information.
Response 1: Revised manuscript line number 18; Line 47-63
Although Abd El-Wahab and El-Shazly [1] recognized the species of thrips as N. variabilis but as Lawrence P. Mound suggested the species may be N. abnormis, a similiarly colored species. So we have removed the presence of soybean thrips in Egypt. The modified sentences are …
“Soybean thrips (Neohydatothrips variabilis [Beach, 1896]) have gained importance as vectors in recent years since the emergence of soybean vein necrosis virus (SVNV) in the United States[2-5], and Canada[6]. Although SVNV presence has been reported in Egypt[1,7] as well, but presence of N. variabilis is still not confirmed. The primary vector of this virus is N. variabilis (Beach) (Thysanoptera: Thripidae), which is prevalent in every soybean-growing state in the United States where SVNV presence has been reported [3-5]. Soybean vein necrosis virus is an important vector and seed transmitted virus which decreases the seed qualitative parameters including oil content and the fatty acid profile [4,8]. With SVNV continuing to affect soybean plants [9], there is a need to find ways to combat the spread of this virus and its vector. Understanding the morphology of the female terminalia of N. variabilis, as well as the egg laying mechanism, could help to develop new strategies for managing both the vector and the disease”.
Point 2: Lines 55-65. This paragraph is irrelevant – UNLESS the author can produce evidence of thelytoky in their target species variabilis.
Response 2: We appreciate the reviewer comments and this paragraph was removed. Line 71 -76 have been removed.
“Other thrips species also exhibit thelytoky, where females develop from unfertilized eggs, and deuterotoky, where both males and females develop from unfertilized eggs. Some species of thrips, such as western flower thrips (Frankliniella occidentalis) have only been reported to exhibit arrhenotoky [10,11], whereas other species of thrips, such as onion thrips (Thrips tabaci) exhibit all three forms of parthenogenesis (viz., thelytoky, arrhenotoky and deuterotoky)”.
Point 3: Lines 66-71 – again irrelevant since no information is provided on the sex ratio of variabilis.
Response 3: These lines were removed. Although we had worked on sex ratio, but it is described in another research paper of life table, so we removed these lines from here.
Line number 80-85
“Exactly how females control the male/female sex ratio in Thysanoptera populations is still not clear. It may be related to the morphology of the female reproductive system, especially the morphology of the spermatheca, which may allow females to control the sex ratio of their offspring, and by extension, the population.”
Point 4: Line 76 – it is not true that female Tubulifera do not have an ovipositor. I find it astonishing that no reference is made anywhere in this mss to the thrips morphological studies of Bruce Heming at Alberta.
Response 4: Line 94
I have replaced the sentence and added “Female Tubulifera, have an eversible chute like ovipositor that lacks teeth [12-15]”.
Heming paper was thoroughly read and reference was added at appropriate places in the manuscript.
Line 88-91 following sentence was added in the manuscript
Heming [16] documented that ovarioles in Thysanoptera are neopanoistic modified from polytrophic ovarioles in Psocopteroid ancestors, which might be a modification for tiny size. Haplo-dipolidy is made possible because male thrips have haploid germline cells.
Line 414-415: The following sentence was added, “Bode [17] while Heming [16] provided a detailed description of evolution of arrhenotoky with reference to germline cells and chromosomes”.
Point 5: Line 86 – Buckman et al (2013) did not recognise two clades of Thripidae – indeed we continue to have difficulty in recognising Thripinae as an independent clade.
Response 5: Line 86 was modified. The modified sentence was
Line 103-106: Phylogenetic studies of Thysanoptera based upon 18S ribosomal DNA, 28S ribosomal DNA sequencing through maximum parsimony, maximum likelihood and Bayesian analysis revealed a monophyletic Tubulifera and Terebrantia [18]. Thrips physapus L., the subject of Bode’s paper, and N. variabilis, our study organism, represent a single clade.
References:
- Abd El-Wahab, A.S.; El-Shazly, M.A. Identification and characterization of soybean vein necrosis virus (SVNV): a newly isolated thrips-borne tospovirus in Egypt. Journal of Virological Sciences 2017, 1, 76-90.
- Zhou, J.; Kantartzi, S.; Wen, R.-H.; Newman, M.; Hajimorad, M.; Rupe, J.; Tzanetakis, I. Molecular characterization of a new tospovirus infecting soybean. Virus genes 2011, 43, 289.
- Bloomingdale, C.; Irizarry, M.D.; Groves, R.L.; Mueller, D.S.; Smith, D.L. Seasonal population dynamics of Thrips (Thysanoptera) in Wisconsin and Iowa soybean fields. Journal of economic entomology 2016, 110, 133-141.
- Irizarry, M. Soybean vein necrosis virus: impacts of infection on yield loss and seed quality and expansion of plant host range. Iowa State University, 2016.
- Keough, S.A. Surveying Indiana Soybean for Soybean vein necrosis virus (SVNV) and Analysis of SVNV on Life History Traits and Host Preference of Thrips Vectors. Purdue University, 2015.
- Bloomingdale Chris; Carl Bradley; Martin Chilvers; Loren Giesler; Russ Groves; Daren Mueller; Damon Smith; Albert Tenuta; Kiersten Wise. Soybean Vein necrosis virus. Crop protection Network 2015.
- Abd El-Wahab, A. Molecular characterization and incidence of new tospovirus: Soybean Vein Necrosis Virus (SVNV) in Egypt. Brazilian Journal of Biology 2021, 84.
- Zhou, J.; Tzanetakis, I.E. Soybean vein necrosis virus: an emerging virus in North America. Virus genes 2019, 55, 12-21.
- Anderson, N.R.; Irizarry, M.D.; Bloomingdale, C.A.; Smith, D.L.; Bradley, C.A.; Delaney, D.P.; Kleczewski, N.M.; Sikora, E.J.; Mueller, D.S.; Wise, K.A. Effect of soybean vein necrosis on yield and seed quality of soybean. Canadian journal of plant pathology 2017, 39, 334-341.
- Brødsgaard, H.F. Frankliniella occidentalis (Thysanoptera: Thripidae)-a new pest in Danish glasshouses. Tidsskrift for planteavl 1989, 93, 83-91.
- Nault, B.A.; Shelton, A.M.; Gangloff-Kaufmann, J.L.; Clark, M.E.; Werren, J.L.; Cabrera-la Rosa, J.C.; Kennedy, G.G. Reproductive modes in onion thrips (Thysanoptera: Thripidae) populations from New York onion fields. Environmental Entomology 2006, 35, 1264-1271.
- Bailey, S.F.; Bailey, S.F. The thrips of California; University of California Press: 1957.
- Stannard, L.J. The thrips, or Thysanoptera, of Illinois. Illinois Natural History Survey Bulletin; v. 029, no. 04 1968.
- Tipping, C. Thrips (thysanoptera). Encyclopedia of Entomology 2008, 4, 3769-3770.
- Mehle, N.; Trdan, S. Traditional and modern methods for the identification of thrips (Thysanoptera) species. Journal of Pest Science 2012, 85, 179-190.
- Heming, B.S. History of the germ line in male and female thrips. In Thrips biology and management, Springer: 1995; pp. 505-535.
- Bode, W. Der Ovipositor und die weiblichen geschlechtswege der Thripiden (Thysanoptera, Terebrantia). Zeitschrift für Morphologie der Tiere 1975, 81, 1-53.
- Buckman, R.S.; Mound, L.A.; Whiting, M.F. Phylogeny of thrips (Insecta: Thysanoptera) based on five molecular loci. Systematic Entomology 2013, 38, 123-133.

Round 2
Reviewer 2 Report
Many thanks - that looks much better from my point of view Accept